# QCD equation of state via the complex Langevin method

Felipe Attanasio[1*], Benjamin Jäger[2] and Felix P.G. Ziegler[2,3]

**1** Institute for Theoretical Physics, Universität Heidelberg, Philosophenweg 16, D-69120, Germany
**2** CP3-Origins & Danish IAS, Department of Mathematics and Computer Science, University of Southern Denmark, Campusvej 55, 5230 Odense M, Denmark
**3** School of Physics and Astronomy, The University of Edinburgh, EH9 3FD Edinburgh, United Kingdom
* pyfelipe@thphys.uni-heidelberg.de

December 4, 2022

## Abstract

**We present lattice simulations on the phase diagram of Quantum Chromodynamics (QCD) with two light quark flavours at finite chemical potential $\mu$. To circumvent the sign problem, we use the complex Langevin method. In this study, we have carried out ab-initio lattice QCD calculations at finite density for a pion mass of $\sim 480$ MeV. We report on the pressure, energy and entropy equations of state, as well as the observation of the Silver Blaze phenomenon.**

## 1 Introduction

Revealing the phase diagram of *Quantum Chromodynamics* (QCD) from first principles is one of the big challenges in modern high-energy physics. Insight into the hot and dense strongly interacting quark-gluon plasma (QGP) phase provides answers to the physics of the early universe and supernovae. Complementary, the cold and dense regions exhibiting a rich structure of hadronic matter contain crucial information for the understanding of neutron stars. Exploring QCD at finite temperature and baryon density is of paramount importance in heavy-ion collision experiments at LHC, RHIC, FAIR, and NICA. On the theory side the lattice formulation of QCD offers a well-established numerical framework to compute the QCD phase diagram from first principles.

The phase structure, including the nature and critical temperature of the chiral and deconfinement transitions, is well understood at vanishing chemical potential [1, 2]. A finite chemical potential, $\mu > 0$, renders the Euclidean action and hence the path-integral measure complex, thus prohibiting the use of conventional Monte Carlo simulations. Moreover, reweighting the phase of the fermion determinant comes with exponential simulation costs as the lattice volume increases. This is the sign problem in lattice QCD. Over the last two decades various solution programs to deal with the sign problem have been established [3]. Among its candidates rank a variety of reweighting methods [4,5], Taylor expansions [6,7] and analytic continuation from imaginary chemical potential [8–11], dual formulations, and the density of states method [12]. Complementary, the complex Langevin (CL) method [13,14] as well as the Lefschetz thimble method and generalizations thereof [15] are based on analytic continuation in the field variables into the non-compact gauge group $SL(3, \mathbb{C})$. A recent overview of lattice efforts to compute the QCD phase diagram can be found in [16].

In this work, we present results on the phase diagram and the extraction of the equation of state of QCD with two mass-degenerate light flavours. In particular, we focus on exploring

the phase diagram in the region of low temperature and finite density. The latter gives rise to a particularly severe sign problem. Our approach to cope with this is the complex Langevin method. Over the last decade, a plethora of tools to guarantee stability and correctness in CL simulations has been developed, see e.g., [17–25]. Here we put those tools to work in QCD with dynamical quarks and low temperatures.

Our CL-based approach allows us to investigate a large portion of the phase diagram, in particular with baryon chemical potentials close to and above the nucleon mass. The focus of this work is on temperatures between 50 MeV and 200 MeV and baryon chemical potential up to twice the nucleon mass, complementing previous studies of the QCD phase diagram [26–32] and the equation of state [5, 7, 33–39]. Our simulations have been performed at fixed volume and lattice spacing, with pions lighter than 500 MeV. We present here the pressure, energy and entropy equations of state (EoS) in the $T - \mu$ plane, and also numerical evidence of the Silver Blaze phenomenon [40]. Other studies of the QCD phase diagram via complex Langevin simulations can be found in [41–45].

## 2   Computational method

Our simulations have been performed in the grand-canonical ensemble by employing the complex Langevin method. This technique allows the circumvention of the sign problem by extending the configuration space for the gauge link variables from the group SU(3) to SL(3, $\mathbb{C}$). Complex Langevin has been successfully used in tackling the sign problem, beyond the cases mentioned above, in the relativistic Bose gas [46], polarised [47] and mass-imbalanced [48] ultracold atoms. For recent reviews, see [14, 49].

We use the standard Wilson plaquette gauge action

$$S_g = \frac{\beta}{3} \sum_x \sum_{\mu < \nu} \mathrm{Tr} \left[ \mathbb{1} - \frac{1}{2} \left( U_{x,\mu\nu} + U_{x,\mu\nu}^{-1} \right) \right], \tag{1}$$

where $U_{x,\mu\nu}$ represents the elementary plaquette at the site $x$ in directions $\mu$ and $\nu$, and $\beta$ is the inverse coupling. The quark contribution is given by the action for $N_f$ flavours of unimproved Wilson fermions

$$S_f = -N_f \, \mathrm{Tr} \log M(U, \mu), \tag{2}$$

with the Dirac operator

$$M_{xy} = (4 + m)\delta_{xy} - \frac{1}{2} \sum_\nu \left[ \Gamma_\nu e^{\mu \delta_{\nu,0}} U_{x,\nu} \delta_{x+\hat{\nu},y} + \Gamma_{-\nu} e^{-\mu \delta_{\nu,0}} U_{x-\hat{\nu},\nu}^{-1} \delta_{x-\hat{\nu},y} \right]. \tag{3}$$

The parameters $m$ and $\mu$ stand for the quark mass and chemical potential, respectively, in lattice units, and $\Gamma_{\pm\nu} = 1 \mp \gamma_\nu$.

Field configurations are generated using the complex Langevin [50, 51] method

$$U_{x,\mu}(\theta + \epsilon) = \exp\left[ i\lambda^a \left( \epsilon K_{x,\mu}^a + \sqrt{\epsilon} \eta_{x,\mu}^a \right) \right] U_{x,\mu}(\theta), \tag{4}$$

where $\eta_{x,\mu}^a$ are white noise fields and the derivative in the Langevin drift, $K_{x,\mu}^a \equiv -D_{x,\mu}^a S$, acts on the group manifold [13, 52]. The step size $\epsilon$ is adaptively changed during the simulation [53], and we make use of the gauge cooling technique [17] to reduce large explorations of the (non-compact) group manifold. Our drift is augmented with the dynamical stabilisation term to help ensuring proximity to the SU(3) manifold [22]. Quantum expectation values are averages for large $\theta$. We have estimated autocorrelation times following [54]. Note that in this work we have employed a modified version of the dynamic stabilisation method, in which

its contribution to the Langevin drift does not use the adaptive step size computed from $K_{x,\mu}^a$. Instead, we keep the step size that multiplies the DS term fixed at $\bar{\epsilon} = 10^{-3}$ in order to ensure that the additional force term depends only on the distance to the unitary manifold. The additional force then has the form $\bar{\epsilon}\,\alpha_{\text{DS}}\,K_{\text{DS}}$. Our CL simulation code is based on the openQCD [55] and openQCD-FASTSUM software packages [56]. The most costly part of the CL simulation is the computation of the fermionic drift force. The quark contribution to the Langevin drift reads

$$\left(K_{x,\mu}^a\right)_{\text{quark}} = N_f \operatorname{Tr}\left[M^{-1}D_{x,\mu}^a M\right]. \tag{5}$$

We have used the even-odd preconditioned conjugate gradient algorithm in [55] applied to the normal equation $M^\dagger M\psi = \eta$ to estimate $M^{-1}$, and compute the trace using the bilinear noise scheme. To further reduce the numerical effort, we update the gauge force more frequently than the fermionic drift. We choose the ratio of gauge over fermion updates to be 16. This needs to be taken into consideration once we conduct the step size extrapolation. As a reference, we performed a finite step size extrapolation for vanishing chemical potential in [57]. Here, results are shown for an average step size of the order of $\mathcal{O}(10^{-3})$.

# 3   Numerical results

We have measured the Polyakov loop,

$$P = \frac{1}{3V}\sum_{\vec{x}}\operatorname{Tr}\left\langle\prod_\tau U_{(\vec{x},\tau),\hat{0}}\right\rangle, \tag{6}$$

where the product is taken along the periodic Euclidean time direction, and quark number density

$$\langle n \rangle = \frac{1}{N_\tau V}\frac{\partial \ln Z}{\partial \mu}. \tag{7}$$

Other thermodynamic quantities, such as pressure, energy, and entropy, are discussed in the next section.

Our simulations have been performed at a fixed lattice spacing of $a \approx 0.06$ fm [58], a spatial volume of $V = 24^3$ in lattice units, and a quark bare hopping parameter of $\kappa = 0.1544$, and inverse coupling $\beta = 5.8$. These input values correspond to pion and nucleon masses of $m_\pi \approx 480$ MeV and $m_N \approx 1.3$ GeV. The value of the stabilisation parameter $\alpha_{\text{DS}}$ has been chosen such that its impact on the observables is minimal [22].

We have scanned the $T - \mu$ plane by varying the quark chemical potential in the range $0 \le a\mu \le 0.28$ in steps of 0.02 and $4 \le N_t \le 64$, corresponding to $0.0 \lesssim \mu \lesssim 920$ MeV and 50 MeV $\lesssim T \lesssim 820$ MeV, respectively. With this type of scan our simulations cover both hadronic and quark-gluon plasma regions, as well as regions of pion and nucleon condensation. Since we work with two degenerate flavors, a finite pion density cannot be observed by increasing the quark chemical potential. However, it is expected that for temperatures below the deconfinement transition a non-vanishing quark density should appear for $\mu \gtrsim m_N/3$. In particular, at $T = 0$ the quark density should vanish for $\mu < m_N/3$. This is known as the *Silver Blaze* phenomenon [40], a peculiar situation where the *absence* of net quark density for $0 \le \mu < m_N/3$ is due to non-trivial cancellation between eigenmodes of the Dirac operator. Information about the deconfinement transition can be obtained from the Polyakov loop, even though it is only an order parameter in purely gluonic theories. A vanishing Polyakov loop indicates the absence of free quarks, while a non-zero value implies the reverse.

We present in fig. 1 the quark number density and in fig. 2 the Polyakov loop as functions of the temperature. The density plot clearly shows direct evidence of the Silver Blaze phenomenon: for $\mu_B > m_N$ the data shows $\langle n \rangle/T^3$ diverging, indicating that $\langle n \rangle$ remains finite as

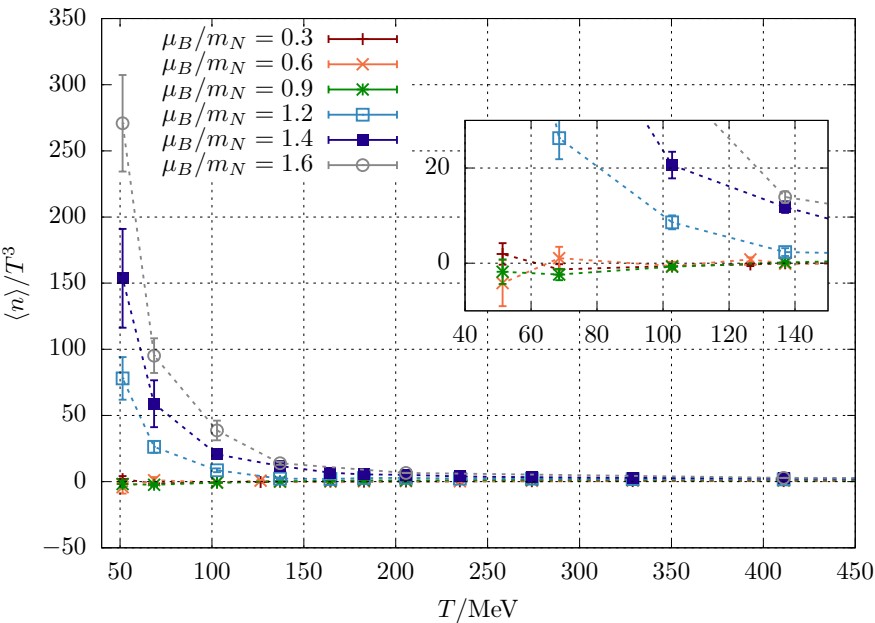

Figure 1: Quark number density as function of temperature for different baryonic chemical potentials. Remnants of the Silver Blaze effect can be seen in the plot. The inset zooms into the lower temperature, low density region and shows that for $\mu_B > m_N$ the density decreases slower than $T^3$ for small temperatures. Dashed lines are to guide the eye. Error bars are statistical only.

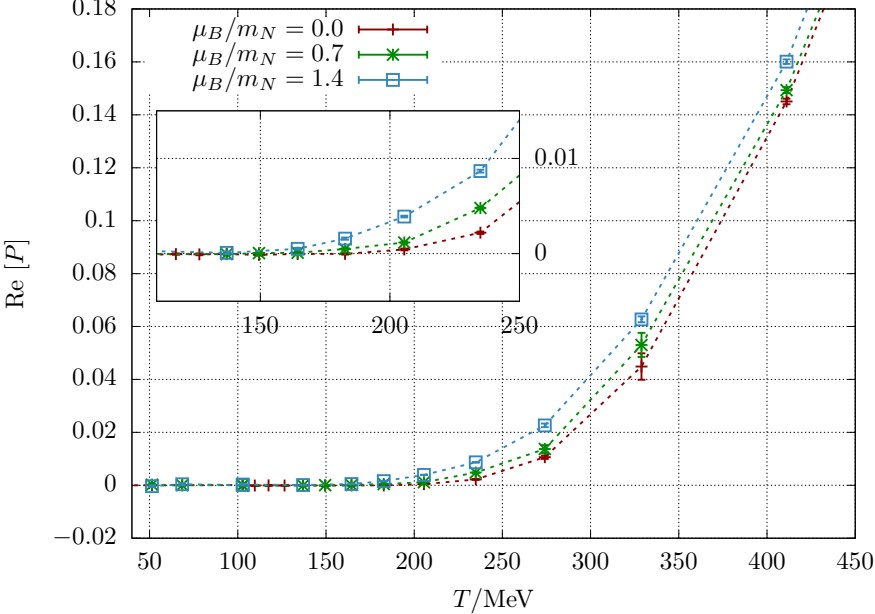

Figure 2: Real part of the Polyakov loop as function of temperature for different baryonic chemical potentials. In the Polyakov loop plot, the inset focuses on the region where $\mathrm{Re}[P]$ starts differing from zero. The Polyakov loop, despite not being an order parameter in QCD, still serves as an indicator of confinement. Dashed lines are to guide the eye. Error bars are statistical only.

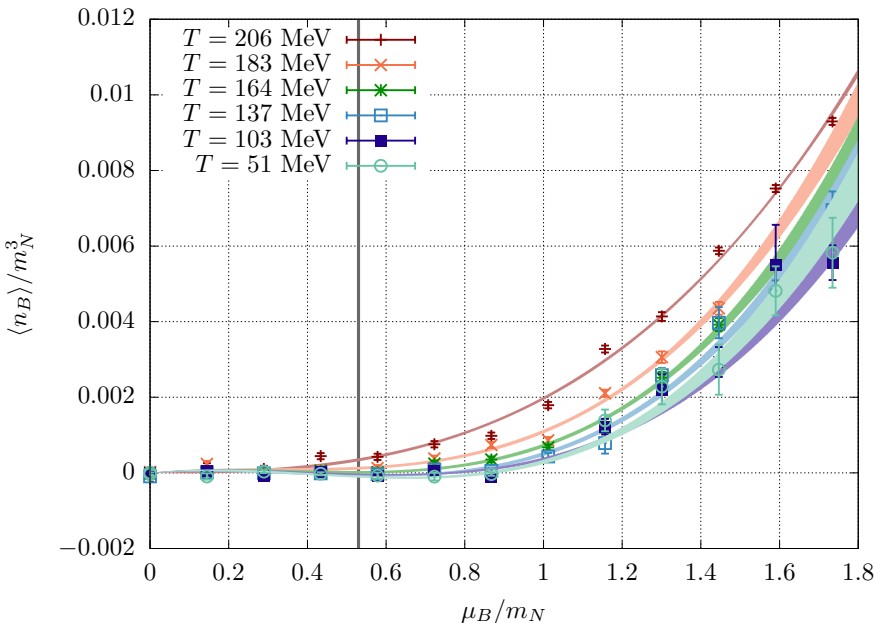

Figure 3: Baryon density normalised by the nucleon mass as function of the baryon chemical potential, including polynomial fits for each temperature. The vertical line indicates $\mu = m_\pi/2$ for reference.

the temperature decreases. In contrast, for $\mu_B < m_N$ the density decreases faster than $T^3$ for low temperatures. Strictly speaking, the Silver Blaze phenomenon only occurs at zero temperature, but the plot shows that for simulations performed below the baryon condensation threshold, $\mu_B < m_N$, the density tends to zero as the temperature decreases, while for $\mu_B > m_N$ it remains finite. The confinement of quarks is indicated by the average Polyakov loop: the figure shows that quarks become free at lower temperatures for larger chemical potentials. This is in qualitative agreement with what has been observed for the QCD phase diagram [31, 35].

## 4 Equation of state

The pressure equation of state can be obtained via

$$\Delta p(\mu_B, T) = \int_0^{\mu_B} d\mu' \langle n(\mu', T) \rangle \tag{8}$$

In order to perform the integration, the density as a function of the chemical potential was fitted by a cubic polynomial for each temperature, shown in fig. 3. The uncertainty on the fit coefficients has been used to compute $1\sigma$ error bands. The choice of a cubic polynomial was inspired by a phenomenological parametrisation of the pressure equation of state for quark matter, see e.g., [59], in terms of a quartic polynomial in $\mu$. Similar studies have been carried out using isospin, rather than baryon, chemical potential, where the viability of compact pion stars [60], and QCD thermodynamics [38] have been investigated.

Using the pressure equation of state we have computed the trace anomaly,

$$\frac{\Delta I}{T^4} = T \frac{\partial}{\partial T} \left[ \frac{p(\mu_B, T)}{T^4} \right] + \frac{\mu_B \, n_B(\mu_B, T)}{T^4}, \tag{9}$$

where $n_B = n/3$ is the baryon number density, and subsequently the energy and entropy

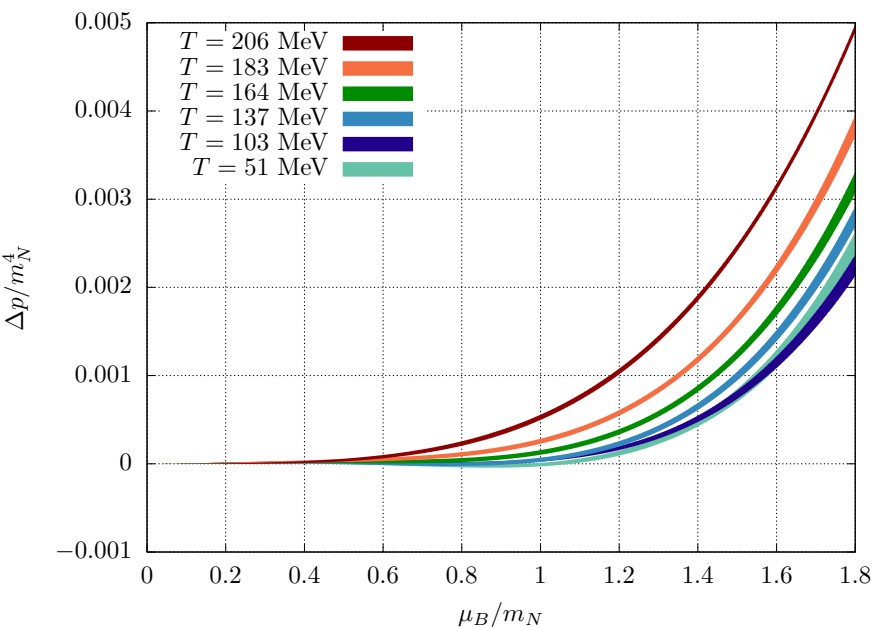

Figure 4: Pressure in units of the nucleon mass as a function of the baryon chemical potential.

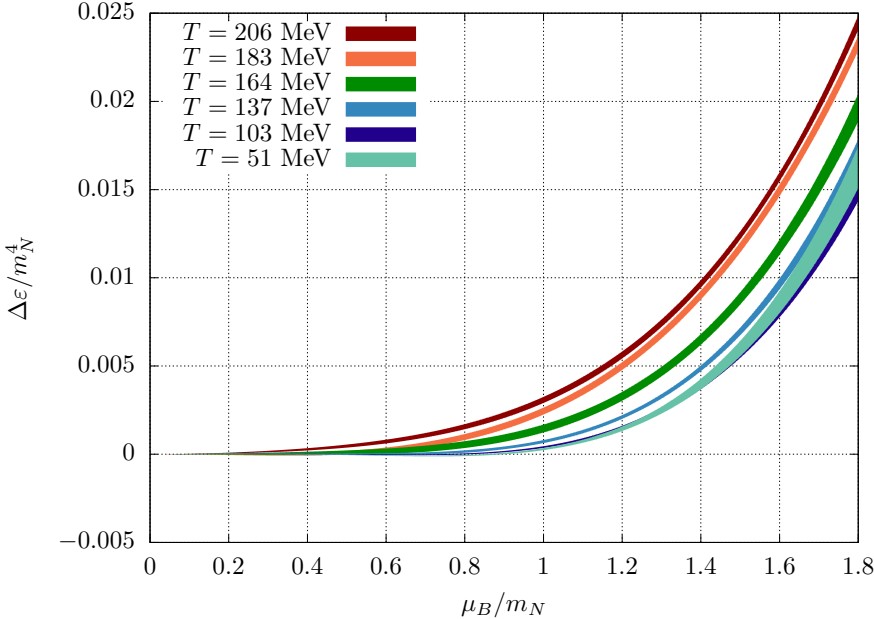

Figure 5: Energy density in units of the nucleon mass as a function of the baryon chemical potential.

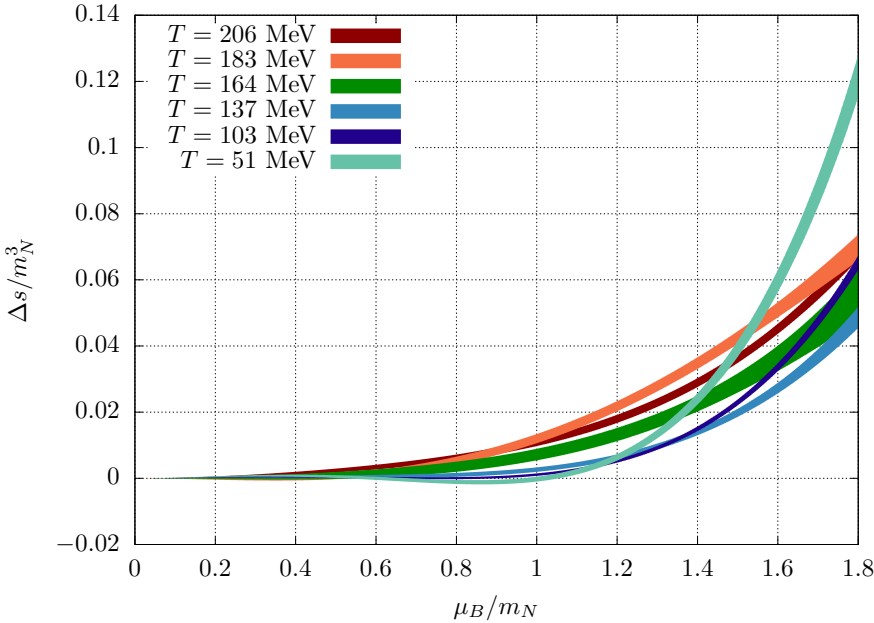

Figure 6: Entropy density in units of the nucleon mass as a function of the baryon chemical potential.

densities,

$$\Delta\epsilon = \Delta I + 3\Delta p \,, \tag{10}$$

$$T\Delta s = \Delta\epsilon + \Delta p - \mu n \,, \tag{11}$$

respectively. The derivative within the trace anomaly has been computed numerically, using the fit coefficients of the pressure computed at each fixed temperature.

The pressure difference with respect to the $\mu_B = 0$ case is displayed in fig. 4, whence a growth can be observed as a function of $\mu_B$. For our lowest temperature, it is noticeable that this growth only starts at $\mu_B = m_N$, since prior to that the baryon number density is essentially zero. The energy and entropy densities can be found in figs. 5 and 6, respectively. An increase in the energy density difference as $\mu_B$ grows is clearly visible for all temperatures considered. In particular, as expected from the nearly vanishing quark number density at low temperatures and chemical potential, $\Delta\varepsilon$ only starts to deviate from zero for $\mu_B \gtrsim m_N$. For higher temperatures, growth starts much sooner.

The stiffness of the equation of state can be inferred from the density as a function of $\mu_B$, shown in fig. 4. It grows slower for lower temperatures, implying that the EoS becomes stiffer as $T$ decreases.

## 5 Summary and outlook

We have presented ab-initio results of the QCD phase diagram with relatively light pions ($\lesssim 480$ MeV) in the $T - \mu$ plane. We cover a baryon number density range of up to $\sim 15$ times the nuclear saturation density $n_0 \simeq 0.16$ fm$^{-3}$ at $T \sim 50$ MeV and up to $\sim 20 n_0$ at $T \sim 200$ MeV. At low temperatures we observe remnants of the Silver Blaze phenomenon, where the quark number density vanishes at $T = 0$ for $\mu < m_N/3$. Our results also show that the pressure equation of state becomes stiffer for lower temperatures. This is in accordance with the behaviour necessary for the stability of neutron stars [61].

In order to further understand dense nuclear matter, our future plans include the addition of the strange quark to our simulations. This would be of particular relevance to studies of neutron stars, as they are known to soften the equation of state. Additionally, our results have been obtained at a finite lattice spacing and need to be continuum and finite step size extrapolated. We plan to employ improved actions to improve the approach to the continuum limit. Another interesting point is the search for the critical end-point (CEP), which requires a fine scan of the phase diagram as well as finite volume scaling analyses and finite step size extrapolation.

## 6   Acknowledgments

We are grateful for discussions with Gert Aarts, Ion-Olimpiu Stamatescu, Jun Nishimura, and Erik Fink. Part of the computing resources for this work were provided by U. of Southern Denmark and DeiC Interactive HPC (UCloud, GenomeDK). Furthermore, this work was carried out using PRACE resources at Hawk (Stuttgart) with project ID 2018194714. This work used the DiRAC Extreme Scaling service at the University of Edinburgh, operated by the Edinburgh Parallel Computing Centre on behalf of the STFC DiRAC HPC Facility (www.dirac.ac.uk). This equipment was funded by BEIS capital funding via STFC capital grant ST/R00238X/1 and STFC DiRAC Operations grant ST/R001006/1. DiRAC is part of the National e-Infrastructure. The work of F.A. was supported by the Deutsche Forschungsgemeinschaft (DFG, German Research Foundation) under Germany's Excellence Strategy EXC2181/1 - 390900948 (the Heidelberg STRUCTURES Excellence Cluster) and under the Collaborative Research Centre SFB 1225 (ISOQUANT).

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
