# Peer review of "QCD equation of state via the complex Langevin method"

_SciPost Physics_

## Round 1 · Referee Report · Anonymous (Referee 1) · 2024-3-31

Strengths

  1. The work contains a numerical demonstration of the Silver Blaze at non-zero baryochemical potential.

  2. The paper estimates the QCD equation of state at temperatures as low as $T \approx 50$MeV, which already overlaps with the region of the phase diagram that is likely probed by neutron star mergers.

Weaknesses

  1. The method of "dynamical stabilisation", which is used in this paper to ensure proximity to the SU(3) manifold after complexification lacks a theoretical argument for its correctness. It is thus (at least at the moment) purely heuristic.

  2. The pion mass used in the simulation is rather heavy at $m_\pi \approx 480$MeV. This is an issue not only for the phenomenological relevance of the study, but also because the severity of the sign problem is much weaker for a heavy quark mass. Furthermore, the two important scales for the Silver Blaze problem, half the pion mass $m_\pi/2$ and a third of the nucleon mass $m_N/3$ are not nearly as well separated as they are in nature.

  3. The work lacks a zero step-size extrapolation, which is a requirement for Langevin type algorithms, as they are not exact (unlike Hybrid Monte Carlo).

  4. There is no continuum extrapolation. Furthermore, the authors use an unimproved Wilson action for the present numerical study. This is known to lead to very large cut-off effects.

  5. The work only uses two dynamical flavours of quarks (u and d), the stange quarks are not included.

Weaknesses 2-5 are explicitly mentioned in the manuscript, and are pointed out as important avenues of future work. I agree. Weakness 1 is more fundamental, and also requires more understanding, but is unfortunately not mentioned in the manuscript.

Report

The manuscript presents a calculation of the QCD equation of state at non-zero baryon density and finite temperatures, using lattice QCD simulations, using a modern incarnation of the complex Langevin method.

One remarkable feature of the paper is that it reaches very low temperatures of around $50$MeV. The other interesting aspect is that the Silver Blaze phenomenon is seen in the numerical data.

The known problems of the complex Langevin method at low temperatures is dealt with via the trick of dynamical stabilization. This trick for the moment lacks a proper theoretical justification. Reading the manuscript, this weakness is not mentioned at all. E.g., the introduction simply states "Over
the last decade, a plethora of tools to guarantee stability
and correctness in CL simulations has been developed... Here we put those tools to work ...".

In my opinion, the paper is mostly interesting to lattice QCD practitioners, since it represents very nice progress in the application of the complex Langevin equation. It is also of some interesting to researches studying dense QCD matter in heavy ion collisions and/or neutron stars, but with many caveats, since the authors are far from the physical value of the QCD parameters (number of flavours, quark masses etc).

I recommend publishing the paper after a few minor changes are made.

Requested changes

Most importantly, please mention somewhere in the manuscript that dynamical stabilization is not guaranteed to give correct results.

I also have a few optional recommendations:

1) Zooming in on Figure 3, it looks like the baryon number goes negative, and similarly, in Figure 4, it looks like $\Delta p$ goes negative at around $\mu_B/m_N \approx 0.8$ for the lowest temperature. I believe this is non-physical behavior, but from Figure 3, it looks to me that it only comes as a result of the polynomial interpolation of the simulation data points, which themselves are consistent with zero. I recommend using a different fit ansatz, where the density cannot go negative.

2) The Polyakov-loop as defined in this work is not UV finite, and is not renormalized in this work, so I recommend calling it the bare Polyakov loop. I won't insist though, since no continuum extrapolation is attempted.

3) Under equation (8), it is mentioned that the analogue of this integral equation was used at non-zero isospin chemical potential to calculate the pressure. However, this integral method was also used recently at non-zero baryochemical potential in 2208.05398 [hep-lat] (at higher temperatures), where the density was directly calculated with phase reweighting. That work is probably also worth mentioning here.

4) To help prospective readers, maybe it is worth specifying some more details of the study in the abstract, such as the fact that it uses a single volume and single lattice spacing, and that it includes two dynamical flavors.

---

## Editorial Decision

awaiting_resubmission